# Metagenomic analysis of the intestinal microbiome reveals the potential mechanism involved in *Bacillus amyloliquefaciens* in treating schistosomiasis japonica in mice

Hao Chen,[1,2] Shuaiqin Huang,[1] Yiming Zhao,[2] Ruizheng Sun,[3] Jingyan Wang,[2] Siqi Yao,[2] Jing Huang,[1,2] Zheng Yu[2]

**ABSTRACT** Schistosomiasis japonica is one of the neglected tropical diseases characterized by chronic hepatic, intestinal granulomatous inflammation and fibrosis, as well as dysbiosis of intestinal microbiome. Previously, the probiotic *Bacillus amyloliquefaciens* has been shown to alleviate the pathological injuries in mice infected with *Schistosoma japonicum* by improving the disturbance of the intestinal microbiota. However, the underlying mechanisms involved in this process remain unclear. In this study, metagenomics sequencing and functional analysis were employed to investigate the differential changes in taxonomic composition and functional genes of the intestinal microbiome in *S. japonicum*-infected mice treated with *B. amyloliquefaciens*. The results revealed that intervention with *B. amyloliquefaciens* altered the taxonomic composition of the intestinal microbiota at the species level in infected mice and significantly increased the abundance of beneficial bacteria. Moreover, the abundance of predicted genes in the intestinal microbiome was also significantly changed, and the abundance of *xfp/xpk* and genes translated to urease was significantly restored. Further analysis showed that *Limosilactobacillus reuteri* was positively correlated with several KEGG Orthology (KO) genes and metabolic reactions, which might play important roles in alleviating the pathological symptoms caused by *S. japonicum* infection, indicating that it has the potential to function as another effective therapeutic agent for schistosomiasis. These data suggested that treatment of murine schistosomiasis japonica by *B. amyloliquefaciens* might be induced by alterations in the taxonomic composition and functional gene of the intestinal microbiome in mice. We hope this study will provide adjuvant strategies and methods for the early prevention and treatment of schistosomiasis japonica.

**IMPORTANCE** Targeted interventions of probiotics on gut microbiome were used to explore the mechanism of alleviating schistosomiasis japonica. Through metagenomic analysis, there were significant changes in the composition of gut microbiota in mice infected with *Schistosoma japonicum* and significant increase in the abundance of beneficial bacteria after the intervention of *Bacillus amyloliquefaciens*. At the same time, the abundance of functional genes was found to change significantly. The abundance of genes related to urease metabolism and *xfp/xpk* related to D-erythrose 4-phosphate production was significantly restored, highlighting the importance of *Limosilactobacillus reuteri* in the recovery and abundance of predicted genes of the gut microbiome. These results indicated potential regulatory mechanism between the gene function of gut microbiome and host immune response. Our research lays the foundation for elucidating the regulatory mechanism of probiotic intervention in alleviating schistosomiasis japonica, and provides potential adjuvant treatment strategies for early prevention and treatment of schistosomiasis japonica.

Address correspondence to Jing Huang, jing_huang@csu.edu.cn, or Zheng Yu, yuzheng@csu.edu.cn.

Hao Chen and Shuaiqin Huang contributed equally to this article. Author order was determined by increasing the alphabetical order of surnames.

The authors declare no conflict of interest.

See the funding table on p. 13.

**KEYWORDS** *Bacillus amyloliquefaciens*, *Schistosoma japonicum*, intestinal microbiome, functional genes

Schistosomiasis, a zoonotic parasitic disease that seriously endangered the health of humans and animals, was mainly prevalent in 78 countries and regions of Asia, Africa, and South America. More than 700 million people worldwide were threatened by the risk of schistosomiasis infection, and almost 240 million people were affected by schistosomiasis, of which 120,000 had severe clinical symptoms. Therefore, schistosomiasis severely damaged human health and greatly hindered social and economic development (1). The main pathogens of human schistosomiasis include *Schistosoma haematobium*, *Schistosoma japonicum*, and *Schistosoma mansoni* (2). Schistosomiasis japonica, which is caused by *S. japonicum*, is currently endemic in China (particularly in the Yangtze River Valley), Indonesia, and the Philippines (3). During the progression of infection with *S. japonicum*, various organs of the host were damaged to various degrees. Allergic dermatitis was caused by the invasion of cercariae from the host skin into the host body. Afterwards, cercariae reached the hepatic portal venous system along with the host's systemic circulatory system, where they developed into adults and caused fibrosis and enlargement of the liver (4). After developing into adults, *S. japonicum* would move in pairs to the mesenteric veins, combine to lay eggs, and thus disrupt the intestinal mucosa to cause intestinal granulation (3). Praziquantel (PZQ) has been the first-choice drug for the clinical treatment of schistosomiasis japonica and has been used worldwide for more than 30 years. Due to its wide range and high frequency of use, schistosomiasis has become less susceptible and has developed resistance to PZQ, which resulted in lower treatment efficacy of PZQ and impeded schistosomiasis control and prevention (5, 6). Based on the current situation of schistosomiasis control, the development and application of novel therapeutic means are crucial.

Probiotic therapy had a long history, and cases of using human feces to treat infections or food poisoning were presented in ancient China (7). Probiotic therapy, which could significantly improve the course of disease exacerbations through the intervention of probiotics, had no adverse effects and no problem of drug resistance (8). After entering the host, probiotics were able to compete for niches through colonization to modulate the microbiota and inhibit adhesion of pathogenic bacteria through the production of bacteriocins, short-chain fatty acids (SCFAs), and biosurfactants (9). They were also able to interact with the host immune system to improve immunity by suppressing proinflammatory cytokines, stimulating secretory IgA production and enhancing communication between the gut and the brain (10). The use of probiotics was also highly recommended in clinics to promote the alleviation of diseases and the repair of organs. For example, the intervention of *Akkermansia muciniphila* in patients with obesity was able to promote the outer membrane protein AMUC_1100 to interact directly with TLR-2, thereby strengthening the intestinal barrier, reducing inflammation, and ultimately improving health (11). Currently, probiotic therapies are also gradually being considered for application in the treatment of schistosomiasis japonica. Based on the recent studies, the intervention of *Bacillus subtilis* significantly alleviated the pathological damage caused by schistosomiasis japonica through modulating the host intestinal microbiome, and the transcriptome analysis revealed significant changes in the expression of genes associated with differentiation of Th1, Th2, and Th17 cells (12). It is indicated that the intestinal microenvironment in the host challenged with *S. japonicum* could be reshaped after the probiotic intervention, but the specific mechanism is still not clear. According to related studies, the intervention of probiotics changed the composition of the microbiota in different parts of the host, thus changing the metabolites of the microbiome, and ultimately effectively alleviating the disease condition by intervening in the host's immune regulatory system (13). Pentanoate, a type of SCFAs metabolically produced by the gut microbiota, could promote glycolysis and activate mTOR activity by inducing the AMPK pathway, which may reduce Th1- and Th2-type immune responses by stimulating secretion of IL-10, ultimately leading to granuloma suppression, but

the specific regulatory mechanisms required further studies (13, 14). In conclusion, microbial metabolites were able to affect the disease course of schistosomiasis japonica by modulating the host's immune response, and probiotic therapy could also be an effective and safe strategy for the prevention and treatment of schistosomiasis.

*Bacillus amyloliquefaciens*, a nonpathogenic and nontoxic probiotic, had highly stable physicochemical properties and could survive in extreme environments and tolerate acid and high temperatures (15–17). In recent years, *B. amyloliquefaciens* has gradually been applied in clinical disease treatment to improve human health. According to relevant studies, it has been found that *B. amyloliquefaciens* could regulate the homeostasis of the intestinal microbiota, downregulate metabolic pathways related to bile acid synthesis, lower levels of inflammatory factors, and restore the repair of the intestinal barrier, ultimately alleviating the symptoms of intestinal inflammatory diseases (18–20). In a previous study, we found that *B. amyloliquefaciens* could reduce the degree of liver fibrosis and intestinal granuloma in *S. japonicum*-infected mice, maintain the homeostasis of the intestinal microenvironment, remodel the intestinal microbiota of *S. japonicum*-infected mice, and modulate the relative abundance of potentially pathogenic bacteria (e.g., *Escherichia-Shigella*) and beneficial bacteria (e.g., *Muribaculaceae*) (21). It was demonstrated that the intervention of *B. amyloliquefaciens* was able to alleviate the pathological condition of schistosomiasis japonica by restoring intestinal homeostasis as well as modulating the relative abundance of beneficial and pathogenic bacteria. However, the mechanisms by which *B. amyloliquefaciens* alleviated the symptoms of *S. japonicum* infection by altering the composition of the microbiota as well as regulating the host's immune response remain unknown and require further investigation.

In this study, shotgun metagenomic sequencing was conducted to analyze the fecal samples collected from the mice acutely infected with *S. japonicum* before and after treatment with *B. amyloliquefaciens*. The changes in composition and gene function of the intestinal microbiome in *S. japonicum*-infected mice after *B. amyloliquefaciens* intervention were analyzed. Based on these results, the potential mechanism by which the intervention of *B. amyloliquefaciens* alleviates the pathological condition of *S. japonicum*-infected mice was proposed. We expect that our findings could provide a safe and effective strategy for the treatment of Schistosomiasis japonica.

## MATERIALS AND METHODS

### Preparation for suspensions of *B. amyloliquefaciens*, mice infection, and sample collection

The detailed protocols about the preparation for suspensions of *B. amyloliquefaciens*, mouse infection, and intragastric administration of *B. amyloliquefaciens* followed the methods reported in our previous research (21). Briefly, *B. amyloliquefaciens* was centrifuged at 8,000 rpm for 10 min at 25°C (Glanlab, Changsha, China). Mice were given 0.3-mL suspension of *B. amyloliquefaciens* every 3 days at a fixed time by intragastric administration according to groups. Specific groups were described as follows: (i) PBS: healthy mice intragastric administrated by phosphate-buffered saline (PBS); (ii) BA: healthy mice intragastric administrated by suspension of *B. amyloliquefaciens*; (iii) SJ: *S. japonicum*-infected mice intragastric administrated by PBS; (iv) SJBA: *S. japonicum*-infected mice intragastric administrated by suspension of *B. amyloliquefaciens*. All the mice were sacrificed by cervical dislocation at day 45 post-infection. The collection of fecal samples and the measurement of weight were conducted 1 day before sacrifice.

### Metagenome DNA extraction and shotgun sequencing

Three samples from each group at day 45 post-infection were selected for shotgun sequencing. Total microbial genomic DNA of fecal samples was extracted following the instructions of OMEGA Soil DNA Kit (D5625-01) (Omega Bio-Tek, Norcross, GA, USA) and was stored at −20°C prior to further assessment. The quantity and quality of

extracted DNAs were measured using a NanoDrop ND-1000 spectrophotometer (Thermo Fisher Scientific, Waltham, MA, USA) and agarose gel electrophoresis, respectively. The extracted microbial DNA was processed to construct metagenome shotgun sequencing libraries with insert sizes of 400 bp by using Illumina TruSeq Nano DNA LT Library Preparation Kit. Each library was sequenced by Illumina HiSeq X-ten platform (Illumina, USA) with PE150 strategy at Personal Biotechnology Co., Ltd. (Shanghai, China).

## Metagenomics analysis

Raw sequencing reads were processed to obtain quality-filtered reads for further analysis. Cutadapt was used for removing sequencing adapters from raw sequencing reads (22). Fastp was used for trimming low-quality reads by using a sliding window algorithm (23). Bowtie2 was used for filtering out the sequence from mice by using the reference genome of mice (mm39) (24). Kraken2 was used for taxonomic classification of the remaining high-quality-filtered reads by using the custom Kraken2 microbiological database, including bacteria and fungi with default settings (25). Bracken was used to estimate the relative abundance of microorganisms in different samples (26). Then, the taxonomic table with relative abundance was used for further analysis. After quality control, the remaining high-quality reads were used for the analysis of the function of genes. Megahit was used for assembling filtered-out reads to contigs (27). Prodigal was used for predicting the coding sequences (CDSs) from the generated contigs (28). CD-HIT was used to remove redundancy from all predicted genes (29). Salmon was used to estimate the read coverage of each gene in different samples, and the abundance of genes normalized by Salmon was used for further analysis (30). eggNOG-mapper was used for function annotation for genes (31). Then, the abundance of the same orthologs (OGs) annotated by eggNOG-mapper was added up, and the OGs table with abundance was used for further analysis. The annotation tables with the abundance about Clusters of Orthologous Genes (COG) and Kyoto Encyclopedia of Genes and Genomes (KEGG) were generated from the result of the annotation from eggNOG-mapper. To determine the host of the predicted genes, Kraken2 was used to align the predicted CDSs to the custom Kraken2 microbiological database, including bacteria. A relatively conservative discrimination method was used to determine the host of genes (genes were considered to be one clade only when genes were determined in the same clades).

## Analysis of species with significant difference in relative abundance

The relative abundance of species was calculated by dividing the reads count of each species by the sum of the reads counts of all species in one sample. The relative abundance was used to perform differential analysis further. The radar plot that was used to show the relative abundance of composition of the intestinal microbiota was performed by the "fmsb" package (32). Species with significant changes in relative abundance were discovered by linear discriminant analysis (LDA) effect size (LEfSe) (http://huttenhower.sph.harvard.edu/galaxy/) (33). The thresholds of *P* value and LDA scores were set at 0.05 and 2.0. The density of distribution in LDA scores was analyzed by the "ggExtra" package (34). Heatmap was used to present the relative abundance of species with significant changes in relative abundance using the "pheatmap" package (35). The relative abundance of species shown in heatmap was normalized by $log_{10}$ calculation. All statistical analyses were performed using the R environment (version 4.2.1) (36). The statistical results were visualized by the "ggplot2" package (37).

## Discrimination of orthologs with significant changes in abundance

A Wilcoxon test was performed to calculate the *P* value for detecting OGs with significant changes in abundance. The Benjamin and Hochberg false discovery rate (FDR) was used to correct the *P* value. The "log2 FoldChange" was explained by *log2 FoldChange* = *log2*[(*A* + *1*)/(*B* + *1*)], where A and B represented the abundance of OGs in different groups, respectively. The thresholds of $P_{FDR}$, *P* value, and absolute value of log2

FoldChange were set at 0.385, 0.05, and 1.0 (38). Significant changes in the abundance of OGs in the four groups were explained by the fact that the trend of significant changes in the abundance of OGs in the SJ group compared to the PBS and BA groups is opposite to that in the SJBA group compared to the SJ group. A Venn diagram using "ggVennDiagram" package was performed to discover OGs with significant changes in abundance, and the definition of significant changes was based on the descriptions above (39). The Sankey plot showed the annotation information about COG of OGs with significant changes in abundance in the SJBA group by using the "networkD3" package (40). All statistical analyses were performed using the R environment. The statistical results were visualized by the "ggplot2" package.

## Detection of KEGG pathway with significant changes in abundance

A Wilcoxon test was performed to calculate the $P$ value for detecting the KEGG pathway with significant changes in abundance. The Benjamin and Hochberg FDR was used to correct the $P$ value. The "log2 FoldChange" was explained by $log2\ FoldChange = log2[(A + 1)/(B + 1)]$, where A and B represented the abundance of the KEGG pathway in different groups, respectively. The thresholds of $P_{FDR}$, $P$ value, and |log2 FoldChange| were set at 0.385, 0.05, and 1.0. The definition of significant changes was the same as that in the previous section. The "UpSetR" package was used to determine the KEGG pathway with significant changes in abundance (41). The "ggalluvial" package was used to show the association between phyla, families, KEGG Orthology (KO) genes, and metabolic reactions (42). Heatmap was used to present the relative abundance of KO genes and KEGG reactions, which was normalized by z-score. The correlation of KO genes, metabolic reaction, and species was shown by a network plot, which was calculated by packages of "Hmisc" and "igraph" and visualized by Gephi software (version 0.10.0) (43–45). The threshold of the $P$ value corrected by FDR and the absolute value of the Spearman correlation were 0.05 and 0.85.

## RESULTS

### Treatment of *B. amyloliquefaciens* significantly changes the composition of intestinal microbiota in mice infected with *S. japonicum*

To explore the impact of *B. amyloliquefaciens* intervention on the intestinal microbiota of mice infected with *S. japonicum*, a radar plot was performed to present the composition of the intestinal microbiota within the four groups. Species that were in the top 10 in relative abundance were used to display. According to the results, *Akkermansia muciniphila*, *Bacteroides caecimuris*, and *Bacteroides* sp. CBA7301 obviously increased in the SJ group; whereas, *Ligilactobacillus murinus*, *Phocaeicola vulgatus*, and *Limosilactobacillus reuteri* obviously increased in the SJBA group compared to the SJ group (Fig. 1A). Then, LEfSe analyses were performed to discover the species with significant changes in relative abundance. LDA scores and $P$ values were calculated by LEfSe. We detected a total of 511 species with significant differences in relative abundance, whose $P$ value was smaller than 0.05 and whose LDA scores were greater than 2.0. The LDA scores were observed to be centrally distributed at 2.0 (Fig. 1B). Subsequently, the relative abundance of species with significant changes in relative abundance was shown by the heatmap. We found that six species showed significant changes in relative abundance among the top 10 species in terms of relative abundance, and the LDA scores of these six species were very high, indicating extremely significant differences among the four groups (Fig. 1C). Finally, the relative abundance of the six species in each group is shown in Fig 1D. These data demonstrated that *B. amyloliquefaciens* treatment significantly changes the composition of the intestinal microbiota in *S. japonicum*-infected mice.

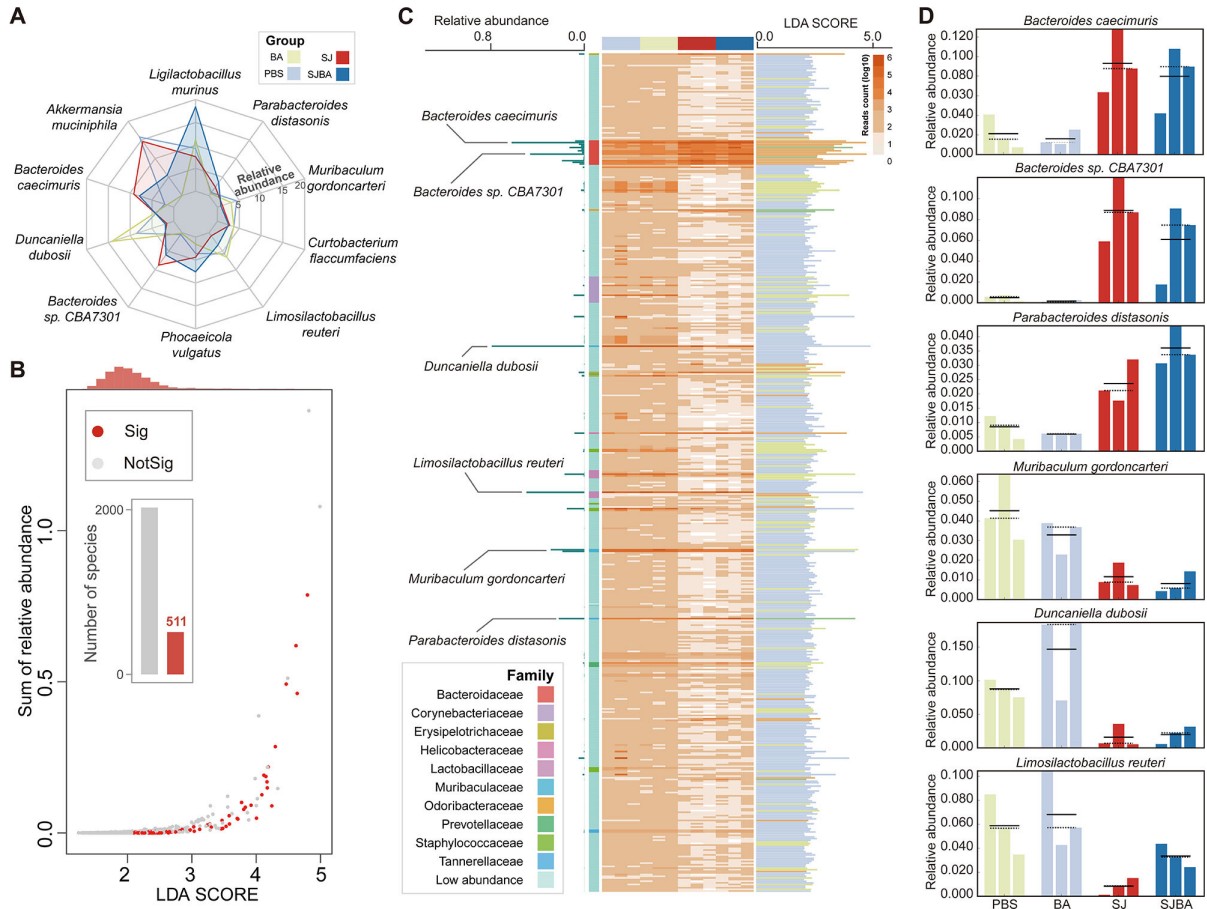

**FIG 1** Alteration in the composition of the intestinal microbiome in the *S. japonicum*-infected mice after treatment with *B. amyloliquefaciens*. (A) Changes in the relative abundance of the top 10 species by relative abundance in each of the four groups. Different colors distinguish the four groups. (B) The scatterplot shows the relationships between relative abundance and LDA scores of the species. The thresholds of the *P* value and LDA scores were set at 0.05 and 2.0. Red points represent species with a significant difference in relative abundance in which the *P* value was less than 0.05 and the LDA scores were greater than 2.0; whereas, gray points mean species did not differ in relative abundance. The red columns in the histogram represent the number of species with a significant difference in relative abundance. The orange column at the top of the plot represents the density of distribution in the LDA scores. "Sig" means significance, and "NotSig" means not significance. (C) The relative abundance and LDA scores of species with a significant difference in relative abundance. The heatmap shows the relative abundance of species with a significant difference in relative abundance. Color depth and relative abundance are proportional. The left of the plot illustrates the relative abundances of species with significant differences in relative abundances among the top 10 species by relative abundance. The right of the plot presents the LDA scores of species with a significant difference in relative abundance. Different colors of columns represent different groups. (D) The right of the plot shows the relative abundance of species with significant differences in relative abundance among the top 10 species by relative abundance in the four groups. Different colors of columns represent different groups.

## Functional gene changes of intestinal microbiome in *S. japonicum*-infected mice after treating with *B. amyloliquefaciens*

The Wilcoxon test and log2 FoldChange were used to determine whether OGs were significantly changed in the SJBA group. After screening, we found a total of 2,055 OGs with significant changes in abundance, of which |log2 FoldChange| was greater than 1.0 (Fig. S1A). Then, a Venn diagram was created to discover the OGs that meet the definition of significant changes (refer to the Materials and Methods). We also found 43 OGs significantly increased in the SJ groups but decreased in the SJBA group, or decreased in the SJ groups but increased in the SJBA group (Fig. 2A). Based on the heatmap about the abundance of 43 OGs, 20 OGs were observed to have a high abundance in the SJBA group (Fig. S1B). To figure out the host of OGs, the highest taxonomic classifications of OGs were shown. As suggested by the results, most OGs were generated by *Bacteroidales*, and most OGs that have a high abundance in the PBS and BA groups were

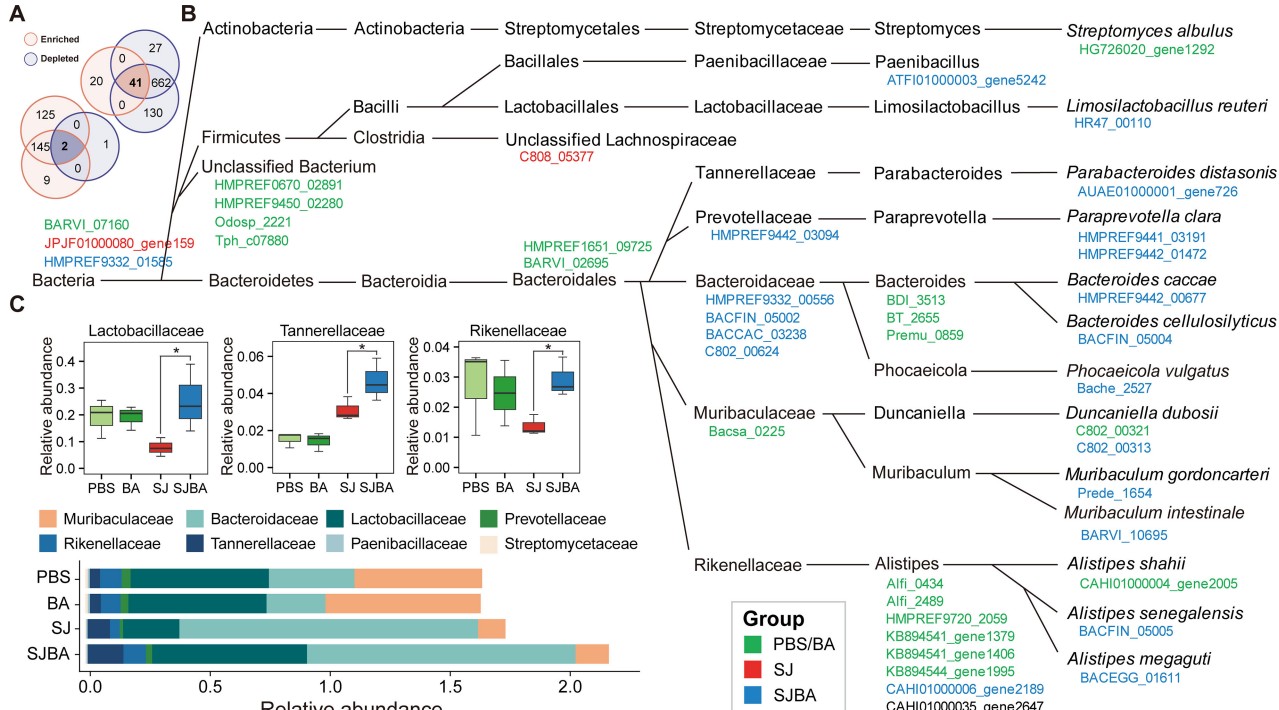

**FIG 2** The taxonomic classification levels of genes with a significant difference in abundance. (A) The Venn diagram shows the number of shared and unique OGs with the opposite trend of significant change in abundance among different groups. Orange represents that OGs were significantly enriched, and purple represents that OGs were significantly depleted. The number of OGs in this plot came from Figure S1. (B) The highest taxonomic classification levels of OGs with a significant difference in abundance. Different colors of OGs represent that OGs were enriched in different groups. (C) The relative abundance of family in the four groups. Data were analyzed by Wilcoxon test. *$P < 0.05$.

observed to be expressed by *Alistipes* (Fig. 2B). We also found eight families expressed most OGs with significant changes in abundance, and Lactobacillaceae, Tannerellaceae, and Rikenellaceae were found to be significantly increased in the SJBA group (Fig. 2C). It meant that they might play important roles in maintaining intestinal homeostasis. Finally, OGs with significant changes in abundance, which had a higher abundance in the SJBA group, were annotated by the COG. The taxonomic classification of OGs was also related to their functions. According to the results, the OG that was annotated as "COG1484 DNA replication protein," "domain of unknown function (DUF 4906)," and "MobA/MobL family" in the COG database had a high abundance in the SJBA group (Fig. 3A). We also defined the host of OGs with significant changes in abundance that had a higher abundance in the SJBA group. We found that *L. reuteri*, *P. vulgatus*, and *Alistipes senegalensis* were significantly increased in the SJBA group compared to the SJ group (Fig. 3B).

## Significant recovery of abundance for the KEGG pathway after treating with *B. amyloliquefaciens* on mice infected with *S. japonicum*

To make it clear which metabolic pathway has significantly changed after treatment with *B. amyloliquefaciens*, differential analysis on the KEGG pathway was performed. Wilcoxon test was performed to detect the metabolic pathways with significant changes among the groups of SJ and PBS, SJ and BA, and SJBA and SJ (Fig. 4A). Then, the significant changes with opposite trends in the comparison of different groups were intersected to obtain the KEGG pathway with a significant difference in abundance that meets the definition of significant change (refer to the Materials and Methods). Based on the results, 27 KEGG pathways with significant changes in abundance were screened (Fig. 4B). Suggested by the information in the KEGG database, the number of KO genes had

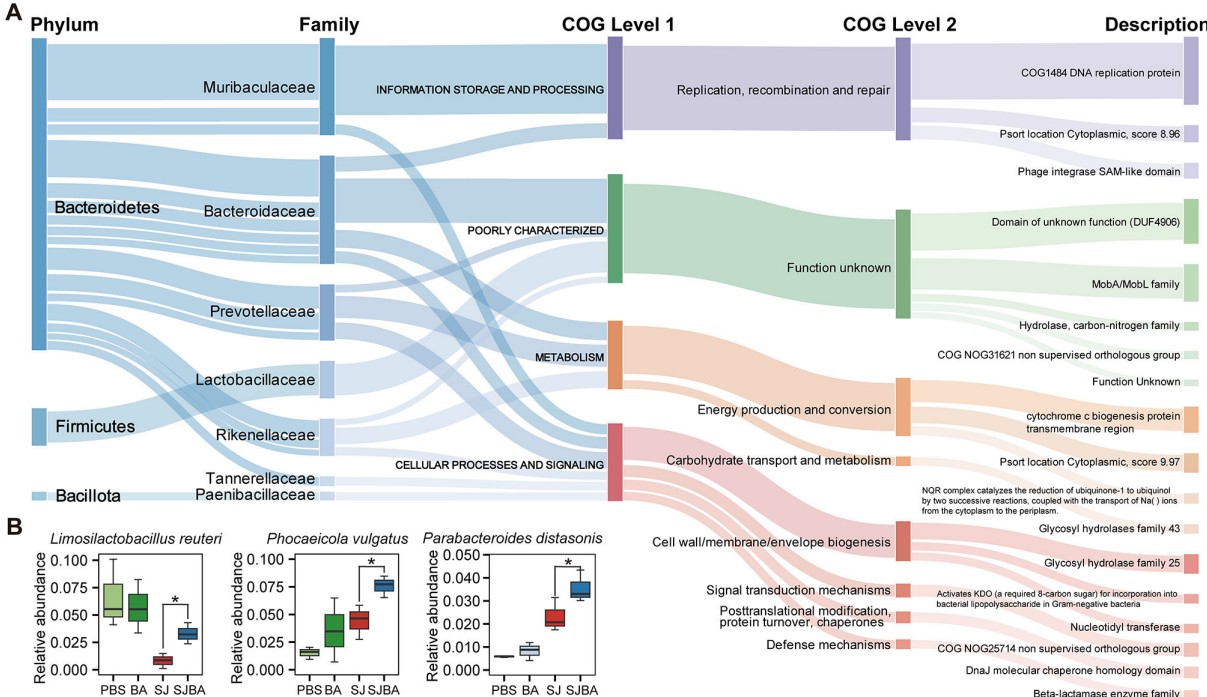

**FIG 3** COG annotations of OGs enriched in the SJBA group. (A) Detailed information of COG annotations of OGs that were enriched in the SJBA group. (B) The relative abundance of species of OGs that had a significant difference in abundance in the SJBA group. Data were analyzed by Wilcoxon test. *P < 0.05.

descriptions that contained the information about enzymes, while enzymes could catalyze metabolic reactions. All the information about enzymes and metabolic reactions came from the KEGG database. At the same time, nucleotide sequences, which could be annotated to KO genes, could also be used for taxonomic annotation. Based on the above associations, KO genes (which came from 27 KEGG pathways with significant changes in abundance and which are involved in the catalysis of metabolic reactions), metabolic reactions, and their information about taxonomic classification that can be annotated were associated. Finally, 8 KO genes, 16 metabolic reactions, and 8 species, which could be correlated based on the association analysis method described above, were obtained (Fig. S2). To determine which KO genes and metabolic reactions changed significantly after treatment with *B. amyloliquefaciens*, Wilcoxon tests were performed. Four KO genes and four metabolic reactions selected by the association analysis were found to be significantly different in the SJBA group compared to the SJ group (Fig. 4C). Six species selected by the association analysis were found to have significant changes in relative abundance based on the results of LEfSe (Fig. 1C). To verify the results of the association analysis, the correlation between four KO genes, four metabolic reactions, and six species was analyzed by Spearman correlation analysis. Based on the results, *L. reuteri* was found to have significant positive correlation with all the KO genes and metabolic reactions, and all the KO genes also had significant positive correlation with all the metabolic reactions. Meanwhile, *B. caecimuris* and *Parabacteroides distasonis* had significant negative correlation with all the KO genes and metabolic reactions (Fig. 4D). Eventually, the relationships among four KO genes and four metabolic reactions were analyzed based on the information from the KEGG database and on the results from Spearman correlation analysis. We found that ureases translated by *ureA*, *ureB*, and *ureC* could catalyze the decomposition of urea into ammonia, and xylulose-5-phosphate/fructose-6-phosphate phosphoketolase (Xfp/Xpk) translated by *xfp/xpk* could catalyze the metabolic reaction to produce D-erythrose 4-phosphate and D-glyceraldehyde 3-phosphate (Fig. 4E).

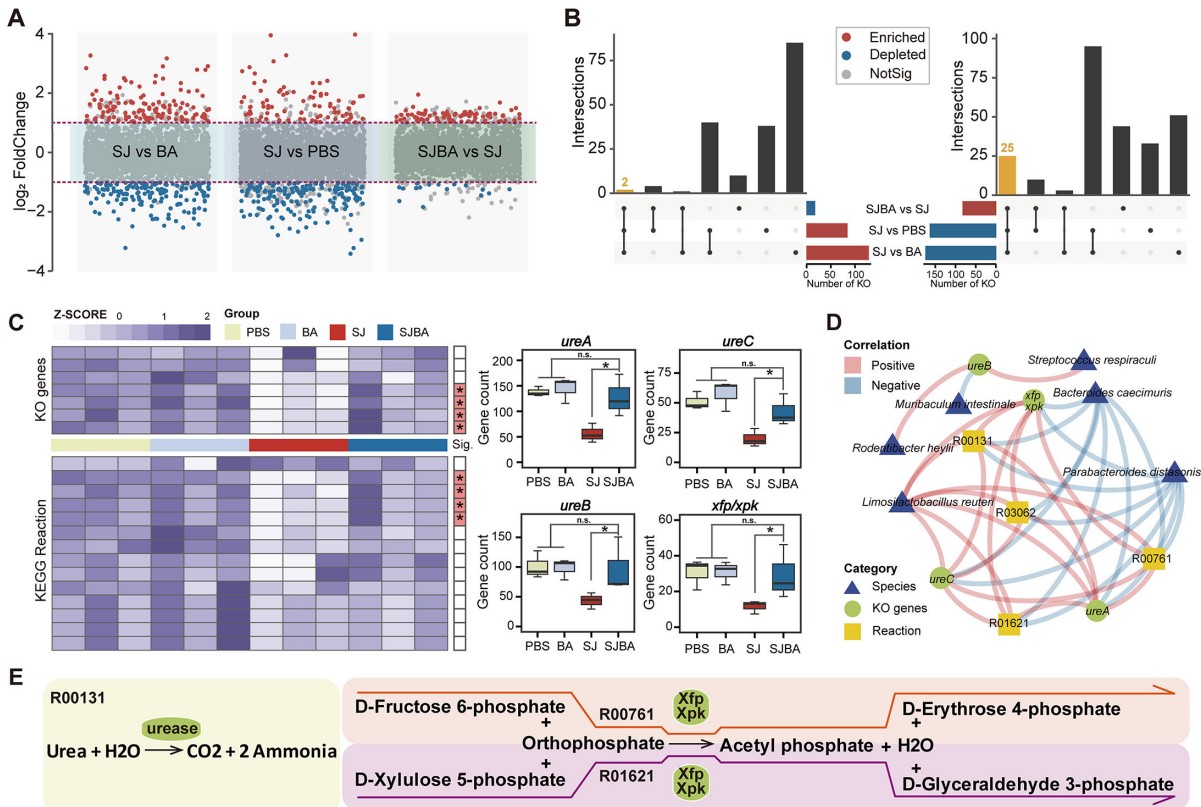

**FIG 4** The relationship between the KEGG pathway and species in the SJBA group. (A) Difference analysis of the KEGG pathway in abundance among different groups. Red points represent significant enrichment of the KEGG pathway in abundance in different groups. Blue points represent significant depletion of the KEGG pathway in abundance. Gray points represent the KEGG pathway in abundance but had no significance. Data were analyzed by the Wilcoxon test. The thresholds of $P$ value and $log_2$ FoldChange were set at 0.05 and 1.0. (B) The UpSet diagram shows the KEGG pathway with significant difference in abundances. Different colors of columns represent different trends of significant change. (C) Screening for KO genes and metabolic reactions with significant differences in abundance. KO genes and metabolic reactions were picked up from Figure S2. The heatmap shows the relative abundance of KO genes and metabolic reactions. The red color on the right of the plot means KO genes or metabolic reactions were significantly different in abundance in the SJBA group. Data were analyzed by the Wilcoxon test. "n.s." means no significance. *$P$ < 0.05. (D) The correlation network between KO genes, metabolic reactions, and species. The positive and negative correlations are separated by the colors red and blue. Triangles in blue indicate species, circles in green indicate KO genes, and squares in yellow indicate metabolic reactions. (E) Metabolic reactions involving KO genes which differed significantly in abundance. The ellipse in green represents KO genes. Different modules in different colors represent different KEGG metabolic reactions.

## DISCUSSION

Schistosomiasis japonica, a parasitic disease mainly endemic in China, seriously threatens human health. After 70-year effort, high-profile achievements have been made in the prevention and treatment of schistosomiasis japonica in China. However, several problems and challenges still remain, such as the increasingly prominent side effects and drug resistance problems of PZQ (46–48). Earlier studies proved that altering the intestinal microbiota of *S. japonicum*-infected mice was able to alleviate the pathological symptoms of schistosomiasis japonica, indicating that it could be a potential therapeutic strategy for schistosomiasis (49). Previously, we found that the pathological injuries in the liver and intestine caused by infection with *S. japonicum* were significantly reduced after *B. amyloliquefaciens* intervention, which was closely associated with the significant recovery of the composition and homeostasis of the intestinal microbiota (21). However, there were no studies elucidating the related mechanisms between the alteration of the intestinal microbiome and the alleviation of symptoms of schistosomiasis japonica in the host. Therefore, in this study, shotgun metagenomic sequencing and functional analysis were performed to evaluate the changes in the taxonomic composition at the species level and gene functions of the intestinal microbiome in *S. japonicum*-infected mice after

*B. amyloliquefaciens* treatment. Additionally, we expected to lay the foundation for the elucidation of the regulatory mechanisms between the intervention of *B. amyloliquefaciens* and the immune regulation of the host in response to *S. japonicum* infection. Our results showed that the intestinal microbiota at the species level of *S. japonicum*-infected mice changed significantly after *B. amyloliquefaciens* intervention, and that the relative abundance of beneficial bacteria increased significantly. Meanwhile, the gene expression of the intestinal microbiome was significantly higher in the SJBA group.

After *S. japonicum* infection, symptoms of intestinal granulomatous disease were induced in the host, which resembled those of inflammatory bowel disease (IBD), typically characterized by hematochezia and unformed stools. At the same time, intestinal granulomas destroyed the intestinal microenvironment and significantly affected the colonizing microbiome of the intestine (2, 50). After *B. amyloliquefaciens* intervention, we found that the relative abundance of bacteria at the species level with the highest abundance was significantly changed in the intestinal microbiome of *S. japonicum*-infected mice. The increasing relative abundance of beneficial bacteria *L. murinus* in the SJBA group was the most obvious, but not significantly different when compared to the SJ group (51). Next, LEfSe analysis identified a significant elevation in the relative abundance of *L. reuteri* and *P. distasonis* in the SJBA group. *L. reuteri* is a well-studied beneficial bacterium that has been applied in various foods and food supplements and has been used as a potential cell factory (52–54). Meanwhile, *L. reuteri* had been clinically proven to have a positive palliative effect on a number of diseases, such as ameliorating diarrhea in children, promoting intestinal epithelial regeneration to repair damaged intestinal mucosa, and promoting the development and function of the regulatory immune system (55–59). In conclusion, *L. reuteri* was important for the promotion of human health, which made it of increasing interest (60). The role of *P. distasonis* in intestinal diseases is mixed. A study of colorectal cancer (CRC) found that IL-10, TGF-β, and tight-junction proteins were significantly higher in the colons of mice that were given *P. distasonis* on a regular basis. This suggests that *P. distasonis* played a protective role in stopping the growth of colon tumors (61). However, *P. distasonis* has been observed to promote colitis in studies of Crohn's disease and ulcerative colitis (UC) (62). Meanwhile, we observed that the abundance of *P. vulgatus* was significantly increased in the SJBA group compared to the SJ group in the analysis of differential genes. *P. vulgatus*, which was capable of disrupting colonic epithelial integrity through the expression of proteases, had been reported to have a potential pathogenic role and could aggravate UC (63–65). However, the prebiotic properties of *P. vulgatus* have also been reported in related studies. *P. vulgatus* had an enzymatic system that degrades complex polysaccharides and was able to form SCFAs (66–68). Currently, there are no relevant studies of *P. distasonis* and *P. vulgatus* in schistosomiasis, so more in-depth studies are needed to explain our findings. There were a lot more *B. caecimuris* and *Bacteroides* sp. CBA7301 in the SJ group, but not many studies have looked at these two bacteria in intestinal diseases.

Therefore, based on the compositional changes of intestinal microbiota at the species level, we found that the abundance of beneficial bacteria in the intestine was significantly upregulated after *B. amyloliquefaciens* intervention, suggesting that restoring beneficial bacteria in abundance due to colonization of *B. amyloliquefaciens* played important roles in alleviating the symptoms of intestinal granulomas caused by *S. japonicum* infection. However, the properties (pathogenic or beneficial) of these bacteria differed significantly in abundance remains to be explored, and their role in schistosomiasis japonica also needs more research to reveal.

Metagenomic shotgun sequencing was not only able to show a more detailed and accurate taxonomic information about the microbiota, but also provided a more relevant information about functional pathways and the abundance of genes at the taxonomic level of all species in the whole microbial community (69–71). Tracing the taxonomic classification of differential genes expressed by intestinal microbiome, we found that *L. reuteri*, *P. vulgatus*, and *P. distasonis* were significantly elevated in the SJBA group,

indicating their active and important roles in the intestinal microbiome. Subsequently, we annotated the identified genes with the KEGG database, and finally we found that the expression of the genes annotated as urease related to urea lysis and *xfp/xpk* was recovered in the SJBA group, and was significantly elevated compared to the SJ group through association analysis and differential analysis. At the same time, under strict validation of Spearman correlation analysis, we found a significantly positive correlation between *L. reuteri* and all genes expressing urease and *xfp/xpk*. Urease can hydrolyze host urea to release ammonia (72). The results of relevant studies demonstrated the pro-inflammatory mechanism of urease, which decomposed urea to produce ammonia, leading to ecological imbalance of the intestinal microbiota and exacerbating the deterioration of colitis (72, 73). These results indicated that the expression of urease genes was not conducive to the recovery of intestinal diseases. However, in this study, we conducted differential analysis on the expression level of urease genes. We found that there were already some genes related to urease expression that exercise normal physiological functions in the host in the control group (PBS and BA groups). At the same time, the expression of genes related to urease in the SJBA group did not have significant differences compared to the control group, but had significantly increased compared to the SJ group. The results illustrated that the intervention of *B. amyloliquefaciens* significantly restored the gene expression related to urease of the intestinal microbiome in *S. japonicum*-infected mice. At the same time, we also conducted taxonomic annotation on the host expressing the gene about urease and found that *L. reuteri* was one of the species that expresses genes related to urease and was the only species with significant differences in abundance. The relevant studies also showed that the supernatant of *L. reuteri* culture medium stopped the growth of pathogenic bacteria *in vitro*, and ammonia was found in the supernatant (74).

Meanwhile, *L. reuteri* has also been shown to be able to produce ammonia (55, 74, 75). Therefore, based on our experimental results, we speculated that the intervention of *B. amyloliquefaciens* promoted significant upregulation of the abundance of *L. reuteri* while restoring the production of ammonia by enhancing the expression of urease genes in *L. reuteri*. The production of ammonia changed the pH value of the intestinal microenvironment, inhibited the colonization of pathogenic bacteria, and ultimately hindered the deterioration of schistosomiasis japonica (76). In parallel, we also observed a significant recovery of abundance of genes, which could be translated to Xfp/Xpk. Moreover, Xfp/Xpk was able to catalyze the generation of D-erythrose 4-phosphate according to the KEGG database. Xfp/Xpk had been reported to be associated with the utilization of prebiotic fiber ingredients, and restoration of their abundance was beneficial for the host to use prebiotic fiber ingredients to restore intestinal health (77). D-erythrose 4-phosphate, one of the products that could be generated by Xfp/Xpk catalysis, was observed to be significantly increased in the mouse model of colitis recovery, suggesting potential relationship between increased levels of D-erythrose 4-phosphate and reduced symptoms of intestinal inflammatory disease (78). Similarly, we traced the microbial host of *xfp/xpk*, and *L. reuteri* was found to be one of the species capable of expressing the *xfp/xpk* and the only one species that differed significantly in abundance. Consulting the KEGG database, we found that D-erythrose 4-phosphate was one precursor molecule for the biosynthesis of aromatic amino acids (e.g., tryptophan, phenylalanine, tyrosine). Also, *L. reuteri* was found to metabolize tryptophan into indole derivatives. Some of these could activate the aryl hydrocarbon receptor (AHR), which could help innate lymphoid cells (ILC) make more IL-22 (79–81). IL-22 could inhibit the colonization of fungi in the intestinal mucosa and enhance the repair of the intestinal mucosa (81). Therefore, the findings suggested potential interaction between the restored abundance of *xfp/xpk* in *L. reuteri* and the host's immune regulatory system, which ultimately alleviates the pathological damage caused by *S. japonicum* infection by improving the strength of the intestinal barrier and strengthening host self-healing of the intestine after injury. However, more detailed and credible regulatory mechanisms involved in this hypothesis needed further studies to be elucidated.

Our results showed significant changes of *L. reuteri* in the composition and abundance of predicted genes of the intestinal microbiome in *S. japonicum*-infected mice, highlighting its importance in maintaining homeostasis of intestinal microbiome. Our results, along with those from other studies, showed that the genes that could be translated to urease and Xfp/Xpk are important in metabolic pathways and in the host immune response to *S. japonicum* infection. However, the exact regulatory mechanisms needed to be elucidated and supported by more experiments. In addition, metabolomic sequencing and immunohistochemical techniques will be used in our future studies, to analyze the differential alterations of physiological metabolites and the expression of inflammatory factors in the intestine and liver of *S. japonicum*-infected mice after *B. amyloliquefaciens* intervention, thereby further elucidating the interactive relationship between the intestinal microbiome and the host immune response, as well as validating our metagenomic results. We are also aware that one limitation of this study is the relatively small sample size in the experimental design. We will expand the number of model mice to increase the credibility and accuracy of the results in the subsequent in-depth analysis. On the other hand, the importance of *L. reuteri* in the gut microbiome of *S. japonicum*-infected mice is highlighted in our results. As one of the colonizing bacteria in the intestine, its ability to secrete reuterin and histamine played important roles in the host's self-protection, repair and reinforcement of the intestinal barrier, and activation of the host's immune response (82, 83). What's more, the abundance of *L. reuteri* was recovered after the intervention of *B. amyloliquefaciens*. If we directly infected the *S. japonicum*-infected mice with *L. reuteri*, would it have better recovery and inhibitory effects on the pathological symptoms caused by the *S. japonicum* infection? Meanwhile, the biosynthesis and metabolism of aromatic amino acids were also one of the mechanisms underlying the regulation of the host immune response, based on our results. At the same time, it has been reported that intervention through the administration of *L. reuteri* to mice accompanied by a tryptophan-rich diet was able to induce the expression of CD4CD8αα T cells$^{++}$ in the intestinal epithelium, which then mediate Th1- and Th2-type immune responses (84). Therefore, intervention with *L. reuteri* treatment and a diet rich in aromatic amino acids might become more effective treatment of schistosomiasis japonica, which would also be further investigated in our subsequent experiments.

In summary, metagenomic analysis showed that the composition of the intestinal microbiota at the species level in *S. japonicum*-infected mice significantly changed after *B. amyloliquefaciens* intervention, and the abundance of intestinal beneficial bacteria significantly increased. At the same time, the abundance of predicted functional genes in the gut microbiome also significantly changed after the intervention. The abundance of genes involved in urease metabolism and Xfp/Xpk related to D-erythrose 4-phosphate production were significantly restored in the SJBA group, highlighting the importance of *L. reuteri* in the recovery and gene expression of the intestinal microbiome. These results also suggested a potential regulatory mechanism between the gene function of the intestinal microbiome and the host immune response. Our study lays the foundation for the elucidation of the regulatory mechanisms of probiotic intervention to alleviate schistosomiasis japonica and provides potential adjuvant therapeutic strategies for the early prevention and treatment of schistosomiasis japonica.

## ACKNOWLEDGMENTS

We would like to express our gratitude to Ying Xiao and Qingqun Wang for their valuable assistance during the process of sacrificing mice. Their contributions were instrumental in ensuring the smooth execution of this essential procedure. We also give our thanks to Umar for his assistance in polishing the language of this manuscript.

This work was funded by the National Natural Science Foundation of China (32170071, 32300051, and 82102428), and the Natural Science Foundation of Hunan Province (2022JJ40663).

J.H. and Z.Y. conceived the study; J.W. isolated and characterized *B. amyloliquefaciens*; H.C., S.H., and S.Y. performed the experiments; H.C. and Y.Z. analyzed the data; H.C. and R.S. visualized the statistical results; H.C. and S.H. wrote the draft; H.C. and S.H. wrote and edited the final manuscript. All authors contributed to the article and approved the submitted version.

## AUTHOR AFFILIATIONS

[1]Department of Parasitology, School of Basic Medical Science, Central South University, Changsha, Hunan, China

[2]Human Microbiome and Health Group, Department of Microbiology, School of Basic Medical Science, Central South University, Changsha, Hunan, China

[3]Department of General Surgery, Xiangya Hospital, Central South University, Changsha, Hunan, China

## AUTHOR ORCIDs

Hao Chen http://orcid.org/0000-0001-8836-368X
Shuaiqin Huang http://orcid.org/0000-0003-1846-7226
Jingyan Wang https://orcid.org/0000-0002-2134-5425
Jing Huang http://orcid.org/0000-0003-4909-766X
Zheng Yu http://orcid.org/0000-0001-7366-3946

## FUNDING

| Funder | Grant(s) | Author(s) |
| --- | --- | --- |
| MOST \| National Natural Science Foundation of China (NSFC) | 32300051 | Jing Huang |
| MOST \| National Natural Science Foundation of China (NSFC) | 32170071 | Zheng Yu |
| MOST \| National Natural Science Foundation of China (NSFC) | 82102428 | Shuaiqin Huang |
| HSTD \| Natural Science Foundation of Hunan Province (湖南省自然科学基金) | 2022JJ40663 | Shuaiqin Huang |

## AUTHOR CONTRIBUTIONS

Hao Chen, Formal analysis, Investigation, Methodology, Validation, Visualization, Writing – original draft, Writing – review and editing | Shuaiqin Huang, Conceptualization, Data curation, Formal analysis, Investigation, software, Supervision, Visualization, Writing – original draft | Yiming Zhao, Formal analysis, Funding acquisition, Methodology, software, Writing – review and editing | Ruizheng Sun, Methodology, Validation, Visualization, Writing – original draft | Jingyan Wang, Conceptualization, Formal analysis, Methodology, Validation | Siqi Yao, Visualization, Writing – original draft | Jing Huang, Conceptualization, Data curation, Formal analysis, Investigation, Methodology, Validation, Visualization, Writing – review and editing | Zheng Yu, Conceptualization, Data curation, Formal analysis, Funding acquisition, Investigation, Methodology, Project administration, Resources, Supervision, Validation, Writing – review and editing

## DATA AVAILABILITY

The datasets analyzed during the current study are available in the Sequence Read Archive (https://www.ncbi.nlm.nih.gov/sra), under accession number PRJNA973216.

## ETHICS APPROVAL

Experiments about the isolation and identification of *B. amyloliquefaciens* were reviewed and approved by the IRB of the School of Basic Medical Science at Central South

University (No. 2021-KT75). All experiments on mice in this study were conducted in strict accordance with the Guide for the Care and Use of Laboratory Animals of the National Institutes of Health. The animal experiments were reviewed and approved by the IRB of the School of Basic Medical Science at Central South University (No. 2021-KT25).

## ADDITIONAL FILES

The following material is available online.

### Supplemental Material

**Supplemental material (Spectrum03735-23-S0001.docx).** Fig. S1 and S2.

### Open Peer Review

**PEER REVIEW HISTORY (review-history.pdf).** An accounting of the reviewer comments and feedback.

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
