## [Reviewer comments · Microbiology Spectrum]

Microbiology Spectrum

Metagenomic analysis of the intestinal microbiome reveals the potential mechanism involved in *Bacillus amyloliquefaciens* treating schistosomiasis japonica in mice

Hao Chen, Shuaiqin Huang, Yiming Zhao, Ruizheng Sun, Jingyan Wang, Siqi Yao, Jing Huang, and Zheng Yu

Corresponding Author(s): Zheng Yu, Central South University

Review Timeline:

Submission Date:	October 20, 2023
Editorial Decision:	January 11, 2024
Revision Received:	January 18, 2024
Accepted:	February 11, 2024

Editor: Patricia Albuquerque

Reviewer(s): The reviewers have opted to remain anonymous.

Transaction Report:

DOI: <https://doi.org/10.1128/spectrum.03735-23>

Re: Spectrum03735-23 (Metagenomic analysis of intestinal microbiota reveals the potential mechanism involved in *Bacillus amyloliquefaciens* treating schistosomiasis japonica in mice)

Dear Prof. Zheng Yu:

Thank you for the privilege of reviewing your work. Below you will find my comments, instructions from the Spectrum editorial office, and the reviewer comments.

Revision Guidelines

Sincerely,
Patricia Albuquerque
Editor
Microbiology Spectrum

Reviewer #1 (Comments for the Author):

The article explored into the intricate relationship between the intestinal microbiota and the therapeutic effects of *Bacillus amyloliquefaciens* in pathologies and diseases caused by schistosomiasis japonica. The following are key concerns in the study: 1. The current study appears to share significant similarities with a previously published article from the same group (Front. Cell. Infect. Microbiol. 13:1172298. doi: 10.3389/fcimb.2023.1172298.). Clarification on the distinctive aspects of the current study

compared to prior work is crucial to claim the novelty and contribution of the research.

2. The article should clarify the methods employed for normalization the load differences among fecal samples for sequencing analysis.

3. How will you exclude the abundance differences of certain kind of microbiomes found in feces are not due to bowel movement changes in groups compared? A non-digestible probe is suggested.

4. When the abundance of certain groups of germs recovered from feces is found to be different between treated and untreated subjects, it is less meaningful than their abundance recovered from gut samples, since the latter represents microbiome colonization more directly. Similar data should be included from gut samples isolated from sacrificed mice.

Reviewer #2 (Comments for the Author):

This study focused on the role of *Bacillus amyloliquefaciens* feeding in alleviating the symptoms of mice infected with *Schistosoma japonicum*. The results showed that the intervention of *B. amyloliquefaciens* increased the abundance of beneficial bacteria in the intestinal tract of infected mice, and changed the functional gene pathways of intestinal microbiome, highlighting the importance of *Limosilactobacillus reuteri*. In general, this study is an interesting scientific research, which clarified the effectiveness of *B. amyloliquefaciens* intervention from the perspective of gene metabolism, and provided potential reference value and basis for the prevention and control of schistosomiasis japonica.

The manuscript is well organized, the methods and results are credible, and the experimental design is appropriate. However, there are still some problems in current version. Therefore, the following points need to be reviewed to improve the clarity and accuracy of the manuscript. After minor revision, this manuscript could be accepted.

The specific issues are as follows:

1. "Microbiome" and "microbiota" were used many times in the manuscript includes titles, yet the meanings of these two words are completely different. It should be used correctly and modified.
2. There are some grammatical errors in the manuscript, please revise and check the English language by a native speaker.
3. Line 20: "the probiotic bacterium" is expressed incorrectly
4. Line 28: "gene expression profile" is expressed incorrectly. Gene expression is generally used for transcriptome analysis. This problem appears many times in the manuscript and needs to be corrected.
5. The writing rules of expression of genes and proteins are different. Attention should be paid to differentiation in the manuscript.
6. Schistosomiasis japonica can be divided into acute stage, chronic stage and late stage. It was observed that the infected mice were sacrificed in the acute phase when the author designed the experiment, which proved that *Bacillus amyloliquefaciens* had a certain protective effect on the infected mice in the acute phase. Does it have a protective effect on infected mice in the chronic phase?
7. In the materials and methods, it is necessary to explain the specific meaning of groups such as "PBS", "BA", "SJ", and "SJBA".
8. The sample size of each group is three, which is a little small. A discussion about this issue should be added to the discussion.
9. Figure 1A, why are the names of these species different in size?
10. What do "Sig" and "NotSig" in Figure 1B mean respectively?
11. Some fonts in figures are too small to be seen clearly. It is better to modify the font size of all figures.
12. Line 363 and 365: "probiotics" is strictly defined and can't be used it casually. Please revise it.

Reviewer #1 (Comments for the Author):

The article explored into the intricate relationship between the intestinal microbiota and the therapeutic effects of *Bacillus amyloliquefaciens* in pathologies and diseases caused by schistosomiasis japonica. The following are key concerns in the study:

Response:

We give our sincere thanks to the reviewer for the critical and detailed comments. According to the professional suggestions of the reviewer, we have made detailed supplement to the method part of the manuscript to reduce the confusion of readers. In addition, as the reviewer said, in order to improve the accuracy of microbiome analysis results, appropriate sampling methods, sample selection and more accurate abundance detection methods are necessary. Therefore, the suggestions of the reviewer are of great help and reference value for our further research. Finally, we hope that the improved manuscript can meet the requirements of reviewer and editor. Thank you again for your valuable comments and suggestions. We will carefully consider them and try our best to improve them.

The specific reply is as follows:

Q1: The current study appears to share significant similarities with a previously published article from the same group (Front. Cell. Infect. Microbiol. 13:1172298. doi: 10.3389/fcimb.2023.1172298.). Clarification on the distinctive aspects of the current study compared to prior work is crucial to claim the novelty and contribution of the research.

Response1: Thanks for pointing out this problem. As you said, the article you mentioned (doi: 10.3389/fcimb.2023.1172298) is one of the previous researches of our team in the early stage, so there are some similarities in the experimental design. Our previous research results have proved that *Bacillus amyloliquefaciens* could well restore disorder of the intestinal microbiota in mice infected with *Schistosoma japonicum*, but the specific mechanism is still unclear. On the basis of previous research, our manuscript observed the changes of intestinal microbiota at species levels and functional genes of mice infected with schistosomiasis japonica after the intervention of *Bacillus amyloliquefaciens* through metagenomic in-depth analysis, in order to better clarify the mechanism of *Bacillus amyloliquefaciens* in alleviating the symptoms of schistosomiasis japonica. Therefore, the analysis of our manuscript from the perspective of functional genes is a further in-depth study comparing to this article (doi: 10.3389/fcimb.2023.1172298). At the same time, we also elaborated the basis of previous research and the problems waiting to be solved in our manuscript. Please see line 20-23: “Previously, the probiotic *Bacillus amyloliquefaciens* has been shown to alleviate the pathological injuries in mice infected with *Schistosoma japonicum* by improving the disturbance of the intestinal microbiota. However, the underlying mechanisms involved in this process remain unclear.”. And line 104-121: “In a previous study, we found that *B. amyloliquefaciens* could reduce the degree of liver fibrosis and intestinal granuloma in *S. japonicum*-infected mice, maintain the homeostasis of the intestinal microenvironment, remodel the intestinal microbiota of *S. japonicum* infected mice, and modulate the relative abundance of potentially pathogenic bacteria (e.g., *Escherichia Shigella*) and beneficial bacteria (e.g., *Muribaculaceae*). It was demonstrated that the intervention of *B. amyloliquefaciens* was able to alleviate the pathological condition of schistosomiasis japonica by restoring intestinal homeostasis as well as modulating the relative abundance of beneficial and pathogenic bacteria. However, the mechanisms by which *B. amyloliquefaciens* alleviated the symptoms of *S. japonicum* infection by

altering the composition of the microbiota as well as regulating the host's immune response remain unknown and require further investigation. In this study, shotgun metagenomic sequencing was conducted to analyze the fecal samples collected from the mice acutely infected with *S. japonicum* before and after treatment with *B. amyloliquefaciens*. The changes in composition and gene function of the intestinal microbiome in *S. japonicum*-infected mice after *B. amyloliquefaciens* intervention were analyzed. Based on these results, the potential mechanism by which the intervention of *B. amyloliquefaciens* alleviates the pathological condition of *S. japonicum*-infected mice were proposed.”

Q2: The article should clarify the methods employed for normalization the load differences among fecal samples for sequencing analysis.

Response2: Thanks for your professional advice. We acknowledged that it was very important to clearly point out the way of data normalization in the manuscript. Therefore, we supplemented the methods about normalization in the relevant analysis in the materials and methods section of the manuscript. Please see line 169: “and abundance of genes normalized by Salmon were used for further analysis”; line 180-183: “The relative abundance of species was calculated by dividing the reads count of each species by the sum of the reads counts of all species in one sample. And relative abundance was used to perform differential analysis further.”; line 190: “Relative abundance of species shown in heatmap was normalized by log₁₀ calculating.”; line 218-219: “Heatmap was used to present the relative abundance of KO genes and KEGG reactions which normalized by z-score.”

Q3: How will you exclude the abundance differences of certain kind of microbiomes found in feces are not due to bowel movement changes in groups compared? A non-digestible probe is suggested.

Response3: Thank you very much for your professional questions. As mentioned in your question, we fully understand your doubts. Relevant researches have also reported that bowel movement can significantly affect the composition of the microbiome (Vujkovic-Cvijin et al. 2020). To solve your question, differences in composition of fecal microbiome between different groups by NMDS analysis based on Bray Curtis distance were performed (refer to the figure below). The results showed that the R-value (0.664) was greater than 0, indicating that the difference of microbiome between different groups was significantly greater than that between samples within the group. This indicated that the differences in the abundance of microbiota were caused by different treatments among different groups, suggesting that the grouping of our experimental design was meaningful. At the same time, we also observed that the microbial composition of the control group (PBS, BA), SJBA group and SJ group differed greatly, indicating that the intervention of *Bacillus amyloliquefaciens* can significantly affect the intestinal microbial composition of mice infected with *Schistosoma japonicum*. Meanwhile, confounding variables was controlled as much as possible to reduce the impact on the results when conducting fecal sample collection. It included mice strain, mice sex, mice age, cercaria infection dose, feeding environment, mice diet, sampling time, stool collection method, stool properties, etc. Of course, we also agreed that using non-digestible probe for detection can improve the accuracy of the results. We will consider incorporating the non-digestible probe into the experimental design in the follow-up researches.

Q4. When the abundance of certain groups of germs recovered from feces is found to be different between treated and untreated subjects, it is less meaningful than their abundance recovered from gut samples, since the latter represents microbiome colonization more directly. Similar data should be included from gut samples isolated from sacrificed mice.

Response4: Thank you very much for your critical suggestions. We think your suggestions are very helpful, and relevant researches also supports your view. Microbial analysis of feces has been accepted as means of determining the relationship between the gut microbiome and host health and disease, because it is believed that feces represent all microbial populations of the entire gut. However, more and more studies have revealed that there are great differences in microbiome between gut and feces recently, including diversity and composition of microbiota, microbial derived metabolites and so on (Martinez-Guryn et al. 2019, Shalon et al. 2023). Different samples will lead to different analysis results. At the same time, some studies have pointed out that the gut microbiome and fecal microbiome represent different meanings. The former could retain signatures of host evolution, while the latter one more reflects the impact of the host's dietary habits on the microbiome (Ingala et al. 2018). According to the previous research (doi: 10.3389/fcimb.2023.1172298), we conducted time series analysis of the fecal microbiome of model mice at different time periods. And this manuscript is an in-depth study of the basis of previous research. So, we selected the fecal microbiome for in-depth metagenomic analysis in order to keep consistent with the basis of previous research and reduce the error caused by changing samples in this manuscript. That's the reason why our manuscript chose to analyze the fecal microbiome of model mice. It is undeniable that microbiome analysis of gut samples in studies could be a better way to reflect the real status of colonization microbiota. Therefore, the microbiome analysis of gut samples will become the focus of our research in the subsequent analysis.

References list

- Ingala, M. R., et al. (2018). Comparing Microbiome Sampling Methods in a Wild Mammal: Fecal and Intestinal Samples Record Different Signals of Host Ecology, Evolution. *Frontiers in Microbiology* 9.
- Martinez-Guryn, K., et al. (2019). Regional Diversity of the Gastrointestinal Microbiome. *Cell Host & Microbe* 26(3): 314-324.
- Shalon, D., et al. (2023). Profiling the human intestinal environment under physiological conditions. *Nature* 617(7961): 581-+.
- Vujkovic-Cvijin, I., et al. (2020). Host variables confound gut microbiota studies of human disease. *Nature* 587(7834): 448-+.

Reviewer #2 (Comments for the Author):

This study focused on the role of *Bacillus amyloliquefaciens* feeding in alleviating the symptoms of mice infected with *Schistosoma japonicum*. The results showed that the intervention of *B. amyloliquefaciens* increased the abundance of beneficial bacteria in the intestinal tract of infected mice, and changed the functional gene pathways of intestinal microbiome, highlighting the importance of *Limosilactobacillus reuteri*. In general, this study is an interesting scientific research, which clarified the effectiveness of *B. amyloliquefaciens* intervention from the perspective of gene metabolism, and provided potential reference value and basis for the prevention and control of schistosomiasis japonica.

The manuscript is well organized, the methods and results are credible, and the experimental design is appropriate. However, there are still some problems in current version. Therefore, the following points need to be reviewed to improve the clarity and accuracy of the manuscript. After minor revision, this manuscript could be accepted.

The specific issues are as follows:

Response: We sincerely thank reviewer for the practical suggestions. Based on these suggestions, we made comprehensive revision of our manuscript. In the manuscript, we have corrected any inappropriate expressions, beautified figures, and polished the language, thus increasing the readability of readers. We sincerely hope that our revised manuscript can satisfy reviewer and editor. Following are our specific modifications.

Q1: “Microbiome” and “microbiota” were used many times in the manuscript includes titles, yet the meanings of these two words are completely different. It should be used correctly and modified.

Response1: Thanks for your advice. We think your advice is very professional and valuable. After learning about the relevant literatures, we learned that microbiota and microbiome do have different meanings. Microbiota refers to microbial ecological groups that study symbiosis or pathology on animals and plants, including bacteria, archaea, protozoa, fungi and viruses. Microbiome refers to the entire habitat, including microorganisms (bacteria, archaea, eukaryotes, and viruses), their genomes (i.e., genes) and the surrounding environmental conditions (Marchesi et al. 2015). We have revised our manuscript to ensure that microbiota and microbiome are used correctly.

Q2: There are some grammatical errors in the manuscript, please revise and check the English language by a native speaker.

Response2: Thanks for your advice. We have invited a native speaker to check the grammar and polish the language expression of our manuscript to enhance the correctness and readability of the manuscript. I hope the revised manuscript can satisfy you.

Q3. Line 20: “the probiotic bacterium” is expressed incorrectly.

Response3: Thanks for your correction. We have changed “the probiotic bacterium” to “the probiotic” in line 20.

Q4: Line 28: “gene expression profile” is expressed incorrectly. Gene expression is generally used for transcriptome analysis. This problem appears many times in the manuscript and needs to be corrected.

Response4: Thanks for your valuable suggestions. As you said, we learned the difference between metagenomics and metatranscriptomics. Metagenomics refers to the collection of genomes and genes from the microbiota through shotgun sequencing, so as to obtain information about the potential functional genes of the microbiota. Metatranscriptomics refers to analysis of the suite of expressed RNAs (meta RNAs) by high-throughput sequencing of the corresponding meta cDNAs. This approach provides information on the regulation and expression profiles of complex microbiomes (Marchesi et al. 2015). We have revised the mistakes in expression and changed the “gene expression profile” to “predicted genes” or “functional genes” to make the manuscript more rigorous.

Q5: The writing rules of expression of genes and proteins are different. Attention should be paid to differentiation in the manuscript.

Response5: Thanks for your correction. After understanding the writing rules, we learned that genes need italics and proteins need capitalization in the manuscript. Therefore, we carefully examined our manuscript and revised it.

Q6: Schistosomiasis japonica can be divided into acute stage, chronic stage and late stage. It was observed that the infected mice were sacrificed in the acute phase when the author designed the experiment, which proved that *Bacillus amyloliquefaciens* had a certain protective effect on the infected mice in the acute phase. Does it have a protective effect on infected mice in the chronic phase?

Response6: Thanks for your question. We think this question is very interesting. According to the current results, *Bacillus amyloliquefaciens* showed good effects in alleviating the symptoms of schistosomiasis japonica in the acute phase. We are also very interested in whether *Bacillus amyloliquefaciens* could show good efficacy against chronic schistosomiasis japonica. This is also next research plans of our team.

Q7: In the materials and methods, it is necessary to explain the specific meaning of groups such as “PBS”, “BA”, “SJ”, and “SJBA”.

Response7: Thanks for pointing out. We agree with your proposal that it’s necessary to clarify the specific meaning of the four groups. We have added detailed descriptions in line 135-141: “Briefly, mice were given 0.3 mL suspension of *B. amyloliquefaciens* every 3 days at fixed time by intragastric administration according to the groups. And specific groups were described as follows: (a) PBS: healthy mice intragastric administrated by phosphate buffered saline (PBS); (b) BA: healthy mice intragastric administrated by suspension of *B. amyloliquefaciens*; (c) SJ: *S. japonicum*-infected mice intragastric administrated by PBS; (d) SJBA: *S. japonicum*-infected mice intragastric administrated by suspension of *B. amyloliquefaciens*.”.

Q8: The sample size of each group is three, which is a little small. A discussion about this issue should be added to the discussion.

Response8: Many thanks to the you for your critical and careful comments. We are acutely aware of the impact of sample size on our findings. We will strengthen the collection of sample size in subsequent studies to improve the reliability and interpretability of the results. We will also further explore and interpret the research data to better support the research results. Finally, we added a discussion of this limitation to our discussion in line 440-444: “At the same time, we also clearly know that there are problems of limited sample size in the experimental design., We will expand the number of model mice to increase the credibility and accuracy of the results in the subsequent in-depth analysis.”.

Q9: Figure 1A, why are the names of these species different in size?

Response9: Thanks for your question. When drawing Figure1A, our original intention was to distinguish difference in the abundance of species by the difference of font size. The larger the font size, the higher the abundance of species. But at present, it seems that such presentation will bring misunderstanding and confusion to readers. So, we decided to change the font size of all species to be the same.

Q10: What do “Sig” and “NotSig” in Figure 1B mean respectively?

Response10: Thanks for your question. This is abbreviations for “Significant” and “Not Significant”. We have added explanations to these two words in Figure legend. Please see line 736: ““Sig” means significance, and “NotSig” means not significance.””.

Q11: Some fonts in figures are too small to be seen clearly. It is better to modify the font size of all figures.

Response11: Thank you very much for your careful review. We have revised all figures and resubmitted the revised version to ensure that all fonts can be clearly displayed and improve the readability of readers.

Q12: Line 363 and 365: “probiotics” is strictly defined and can't be used it casually. Please revise it.

Response12: Thank you very much for your professional advice. We have made modifications in our manuscript, changing “probiotics” to “intestinal beneficial bacteria”.

Reference list:

Marchesi, J. R., et al. (2015). The vocabulary of microbiome research: a proposal. *Microbiome* 3.

Dear Patricia Albuquerque,

We thank the editor and all reviewers for their comments and suggestions. We agree that our manuscript still needs improvement. Therefore, we have made complete revisions of our manuscript, including the improvement of the method part, the beautification of figures, and the supplement of discussion. We are very grateful to all reviewers for their constructive and professional suggestions. At the same time, we also invited a native speaker to check and improve the language expression of our manuscript, which improved the readability of the manuscript.

Finally, we sincerely hope that our revised manuscript can satisfy you. We hope that the editor will review and evaluate our manuscript. We would be grateful if our manuscript could be accepted.

Thank you and best regards.

Yours sincerely,

Zheng Yu

Re: Spectrum03735-23R1 (Metagenomic analysis of the intestinal microbiome reveals the potential mechanism involved in *Bacillus amyloliquefaciens* treating schistosomiasis japonica in mice)

Dear Prof. Zheng Yu:

Your manuscript has been accepted, and I am forwarding it to the ASM production staff for publication. Your paper will first be checked to make sure all elements meet the technical requirements. ASM staff will contact you if anything needs to be revised before copyediting and production can begin. Otherwise, you will be notified when your proofs are ready to be viewed.

Sincerely,
Patricia Albuquerque
Editor
Microbiology Spectrum

Reviewer #1 (Public repository details (Required)):

sequencing data

Reviewer #1 (Comments for the Author):

"To solve your question, differences in composition of fecal microbiome between different groups by NMDS analysis based on Bray Curtis distance were performed (refer to the figure below). The results showed that the R-value (0.664) was greater than 0, indicating that the difference of microbiome between different groups was significantly greater than that between samples within the group."

I don't see how this can resolve my concern. Fecal amount may be measured and compared to indirectly exclude the possible causes of bowel movement.

Reviewer #2 (Public repository details (Required)):

accession number PRJNA973216

<https://www.ncbi.nlm.nih.gov/bioproject/PRJNA973216/>

Reviewer #2 (Comments for the Author):

I'm happy with the revision by authors.